# TOWARDS PIXEL-LEVEL MLLM PERCEPTION VIA SIMPLE POINTS PREDICTION

## ABSTRACT

We present **SimpleSeg**, a strikingly simple yet highly effective approach to endow Multimodal Large Language Models (MLLMs) with native pixel-level perception. Our method reframes segmentation as a simple sequence generation problem: the model directly predicts a **sequence of points** (textual coordinates) delineating object boundaries, entirely within its language space. To achieve high fidelity, we introduce a two-stage SFT→RL training pipeline, where Reinforcement Learning with an IoU-based reward refines the point sequences to accurately match ground-truth contours. We find that *the standard MLLM architecture possesses a strong, inherent capacity for low-level perception* that can be unlocked without any specialized architecture. On segmentation benchmarks, SimpleSeg achieves performance that is comparable to, and often surpasses, methods relying on complex, task-specific designs. This work lays out that precise spatial understanding can emerge from simple point prediction, challenging the prevailing need for auxiliary components and paving the way for more unified and capable MLLMs. Code, data and model are publicly accessible at https://github.com/simpleseganonymous/SimpleSeg.

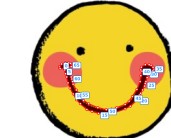

Around 70 points

It is at [[[0.819, 0.453], [0.779, 0.453], [0.752, 0.475], [0.752, 0.487], [0.760, 0.497], [0.760, 0.500], [0.752, 0.538], [0.745, 0.569], [0.728, 0.618], [0.708, 0.662], [0.694, 0.684], [0.670, 0.708], [0.662, 0.718 ], ⋯⋯ , [0.752, 0.650], [0.775, 0.603], [0.781, 0.585], [0.786, 0.564], [0.794, 0.539], [0.797, 0.517], [0.800, 0.514], [0.802, 0.496], [0.806, 0.494], [0.819, 0.494], [0.833, 0.490], [0.838, 0.481]]]

**SimpleSeg: Multimodal Large Language Model**

Where is the mouth?

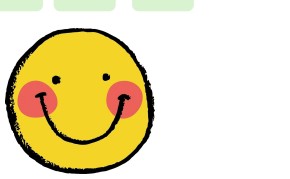

Figure 1: In this work, we explore the limits of MLLM pixel-level perception by predicting the next point in a contour with the simplest approach possible. Without introducing any complex architectures or special patterns, we show how even minimalistic point prediction can achieve effective segmentation at the pixel level.

# 1 INTRODUCTION

Multimodal Large Language Models (MLLMs) have rapidly advanced open-ended vision–language understanding, delivering strong performance across captioning, VQA, and interactive grounding (Liu et al., 2024; OpenAI, 2024; Comanici et al., 2025).

Yet, despite impressive semantic competence, today's Multimodal Large Language Models (MLLMs) remain largely *image-level* in their perception, struggling to precisely localize and delineate fine structures—from object boundaries to thin parts—that are essential for genuine spatial understanding. This limitation is partly rooted in the evolution of multimodal foundational models. While perception is a cornerstone of multimodal tasks, dense prediction tasks like segmentation have historically been overlooked as a foundational capability, as they often rely on specialized decoders or complex architectural designs not native to language-centric models. In contrast, object grounding and detection have been widely adopted, largely because bounding boxes can be conveniently represented as plain text coordinates (e.g., $x_1, y_1, x_2, y_2$) and easily integrated into the pre-training pipeline. However, we argue that the coarse localization offered by bounding boxes is insufficient for the next generation of applications. Such pixel-level grounding is not merely cosmetic: it is foundational for controllable image editing (Shi et al., 2024), vision-based tool use (Wang et al., 2025), and GUI-grounded agents (Liu et al., 2025b; Qin et al., 2025) that must reason, act, and communicate about precisely *where* things are. Therefore, to bridge this gap, we move beyond coarse bounding boxes and consider a more precise and granular approach: point prediction.

> **Takeaway1** Standard MLLM Architectures have a strong, inherent, but previously latent, capacity for precise, pixel-level perception.

A prevalent approach augments MLLMs with task-specific decoders (e.g., SAM- or RPN-style heads) on top of the multimodal backbone (Lai et al., 2024; Zhang et al., 2024b; He et al., 2024; Ren et al., 2024; Rasheed et al., 2024; Zhang et al., 2023; Wu et al., 2024; Jiannan et al., 2024). While effective, this design couples architecture to specific tasks, complicates end-to-end training with extra parameters, and pushes outputs out of the language space, weakening interpretability and compositional reasoning. As a result, fine-grained perception remains underexplored as a core capability of native MLLMs. Decoder-free methods, such as Text4Seg (Lan et al., 2024), serialize masks as text, but suffer from dense token budgets and compromised interpretability. VisionLLM (Wang et al., 2024) emits polygons but restricts them to a small number of vertices, limiting its performance. Both methods fail to deliver pixel-level segmentation with the generality and reasoning fluency of modern MLLMs.

In this work, we investigate a strikingly simple question: can an MLLM achieve high-fidelity segmentation by merely predicting a sequence of points? We present **SimpleSeg**, a minimalist decoder-free approach that reframes segmentation as simple, sequential point prediction entirely within the language space. More than just a method, our work serves as a crucial finding: ***we demonstrate that standard MLLM architectures possess a strong, inherent capacity for fine-grained perception***, a potential that can be unlocked without any specialized decoders or complex output formats. This approach preserves the model's generalist architecture, dramatically simplifies the training pipeline, and naturally unifies object localization tasks (points, boxes, and masks) under a single, human-readable textual interface.

Specifically, we first introduce a systematic point-sequence-based representation for segmentation masks that efficiently scales data preparation. Based on this, we generalize the perceptual localization task beyond text queries: any target can be an input or output in a 4-tuple, $[text, point, box, mask]$, allowing for a rich combination of task formats that boosts data efficiency and robustness.

To make this simple point prediction effective, we design a two-stage SFT→RL training pipeline. After a standard supervised fine-tuning (SFT) stage to learn the basic task format, we pioneer the use of Reinforcement Learning (RL) to optimize the entire generated sequence of points. By using an IoU-based reward, RL directly refines the fidelity and closure of the resulting shape without altering the MLLM's architecture. To our knowledge, this is the first work to successfully apply reinforcement learning to a decoder-free MLLM for segmentation.

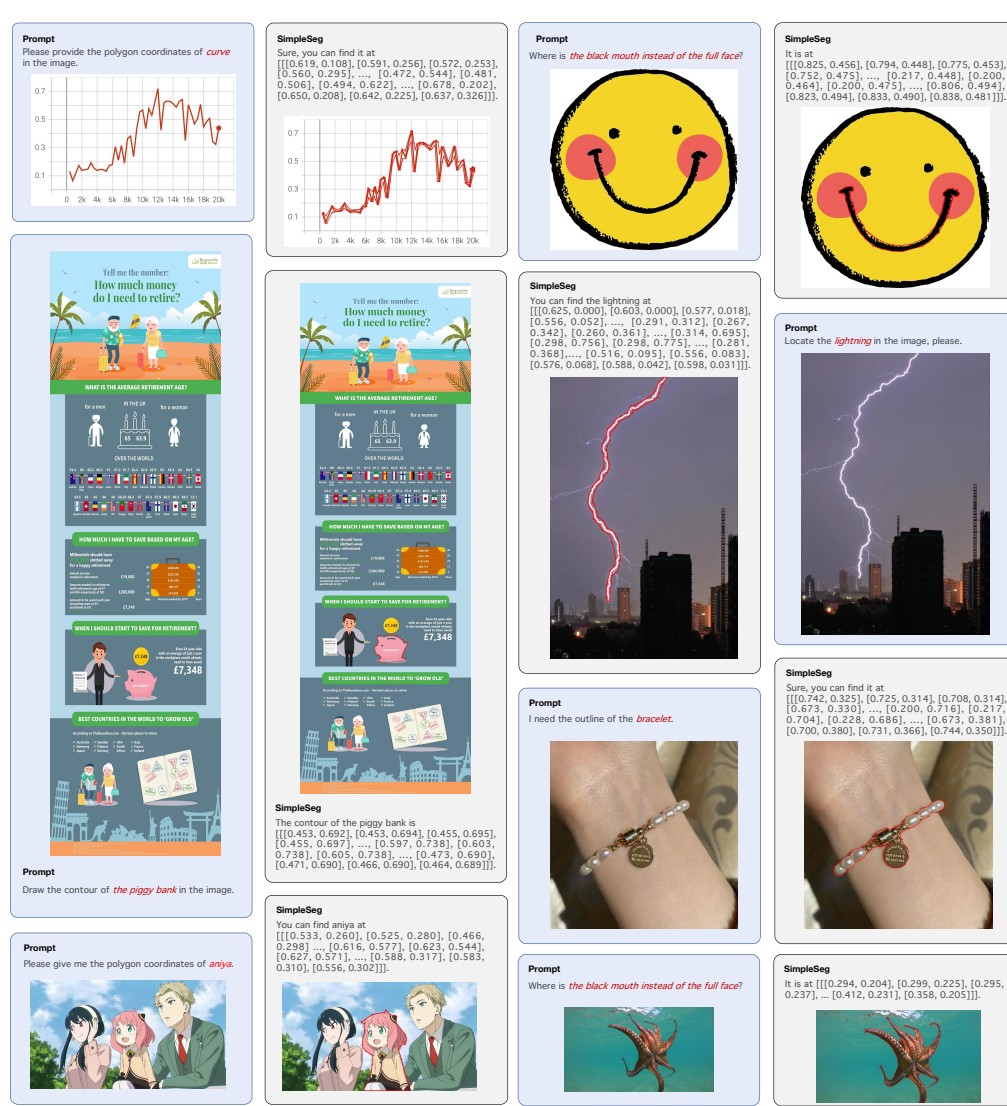

Figure 2: Segmentation results of SimpleSeg on natural and non-natural images. These examples highlight the model's excellent generalization, showing its precise pixel-level perception is not confined to real-world objects. The model successfully segments targets from natural photographs (the lightning) and performs with equal precision on various forms of "in-screen" or digitally generated content, including anime, data charts, and infographics.

**Takeaway2** Powered by its native, language-aligned output format, SimpleSeg shows precise pixel-level perception beyond real-world objects and strong generalization of on natural and non-natural images, highlighting its potential as a core capability for generalist Vision-Language Models.

Empirically, our model attains high-quality, native fine-grained perception and generalizes robustly across diverse domains and resolutions, as illustrated in Fig. 2. On challenging referring benchmarks such as the refCOCO series, SimpleSeg achieves performance that is comparable to, and often surpasses, prominent methods that rely on complex, task-specific decoders. The main contributions can be summarized in three folds as follows: The main contributions can be summarized as follows:

- We present a minimalist, decoder-free approach for MLLM segmentation based on simple point sequence prediction, challenging the necessity of complex architectural additions.

- We provide a key finding that standard MLLM architectures possess a strong, inherent potential for pixel-level perception, which can be unlocked with the right training methodology.

- We are the first to propose and validate an SFT→RL pipeline for this task, using sequence-level IoU rewards to directly optimize the quality of the generated geometry.

- We demonstrate that our simple approach achieves performance comparable to or exceeding that of more complex decoder-based systems on standard referring segmentation benchmarks.

In addition to the above, SimpleSeg offers several key benefits:

- **Simplicity**: SimpleSeg requires no specialized modules and adheres to the standard MLLM architecture, it can be seamlessly and efficiently integrated as a new, core pre-training task for foundation models, similar to visual grounding.

- **Task Generality**: By framing segmentation as a text-generation problem, our approach is inherently flexible. The model can be easily adapted to a wide range of vision-language tasks that require precise spatial localization.

- **Interpretable Output**: The model generates explicit, human-readable coordinate sequences instead of dense pixel masks. This transparency simplifies debugging and makes the output directly usable for downstream applications like interactive editing or tool use.

This makes SimpleSeg not just an efficient solution, but a versatile framework for deploying pixel-level perception in multimodal models with a broad range of applications.

## 2 RELATED WORK

**Multimodal Large Language Models.** Multimodal Large Language Models (MLLMs) have significantly advanced vision-language tasks by extending the reasoning capabilities of LLMs to the visual domain (Yin et al., 2024). Early pioneering models like LLaVA (Liu et al., 2024) established a strong foundation for multimodal instruction following. Subsequent works (Lu et al., 2024; OpenAI, 2024; Comanici et al., 2025) have further pushed the boundaries of performance by scaling up model and data size. However, a common limitation persists: the perception of these models is typically coarse and image-level. They excel at high-level description and reasoning but lack the native ability for precise, pixel-level localization, which remains a largely underexplored frontier.

**Approaches to Pixel-Level MLLM Perception.** Efforts to equip MLLMs with dense, pixel-level perception have primarily followed two distinct paths, creating a central dilemma between performance and architectural integrity. The first, a **hybrid approach**, augments a general MLLM backbone with specialized, task-specific decoders (Lai et al., 2024; Zhang et al., 2024b; Ren et al., 2024; Jiannan et al., 2024; Rasheed et al., 2024; Xia et al., 2024; Zhang et al., 2024a). These external modules, often inspired by SAM or other segmentation architectures, can achieve strong performance on specific tasks like referring segmentation. However, this modular design comes at the cost of architectural purity; it introduces extra parameters, complicates training, and moves the final output outside the native language space, undermining the vision of a truly unified multimodal model.

The second, a **unified serialization approach**, attempts to keep all outputs within the language space by representing masks as text sequences. Early methods explored formats like run-length encoding (RLE) (Lan et al., 2024) or coarse polygons with a few vertices (VisionLLM (Wang et al., 2024)). While conceptually aligned with the end-to-end philosophy of LLMs, these serialization techniques have so far struggled to achieve high fidelity, often suffering from excessive token consumption, low resolution, or an inability to capture fine-grained detail. Consequently, achieving high-performance, pixel-level perception *natively* within a standard MLLM architecture—without sacrificing either simplicity or precision—remains a fundamental open challenge.

## 3 METHODOLOGY

We present **SimpleSeg**, a simple yet effective framework that equips a vanilla MLLM with native pixel-level perception *via simple points prediction*. The key idea is to keep segmentation entirely inside the language space by predicting a *point trajectory* (i.e., an explicit sequence of 2D coordinates) that traces the target contour. This design is decoder-free, architecture-agnostic, and naturally unifies points, boxes, and masks under one textual interface.

Figure 3: Overview of our data annotation pipeline, which incorporates modules for object detection, mask segmentation, points conversion, and instance caption.

## 3.1 TASK FORMULATION AND DATA CONSTRUCTION

**Outputs as Text in One Space.** Perceptual location information commonly appears as (i) a center `point`, (ii) a `bbox`, or (iii) a `mask`. Keeping all outputs as *text tokens* preserves the MLLM's generalist interface and enables direct composition with language prompts and tools. We therefore adopt the following normalized, human-aligned formats:

$$\texttt{<point>}: [x, y]$$
$$\texttt{<bbox>}: [x_1, y_1, x_2, y_2]$$
$$\texttt{<mask>}: [[x_1, y_1], [x_2, y_2], \ldots, [x_V, y_V]]$$

where coordinates are normalized to $[0, 1]$ and $V$ is variable.

**Masks as Point Trajectories (Contours).** Instead of dense per-pixel encodings (e.g., R-RLE (Lan et al., 2024)), we represent a mask by an *explicit point trajectory* that sparsely samples its boundary. This brings three benefits: (1) **interpretability** (human-readable coordinates), (2) **compositionality** (same token space as text/points/boxes), and (3) **controllable token budget** (linear in the number of vertices rather than image resolution). For training data derived from binary masks, we extract polygonal contours using the Suzuki–Abe algorithm (Suzuki et al., 1985) (OpenCV (Itseez, 2015)), enforce a consistent traversal order (clockwise), and optionally apply a tolerance-based sparsification to obtain a compact simple points sequence.

**A Unified Query Interface.** Following the spirit of promptable segmentation, we model grounding over the tuple

$$\texttt{target} = [\texttt{text}, \texttt{point}, \texttt{bbox}, \texttt{mask}],$$

and instantiate *queries* as Cartesian products of the available elements (e.g., (text→bbox), (point→mask)). Two examples are:

**Q:** What is the bounding box of `<text>`?  **A:** `<bbox>`.

**Q:** Give the polygon of the object at `<point>`?  **A:** `<mask>`.

This interface (i) multiplies supervision sources by recombining weak labels (e.g., points/boxes from masks via min/max or centroid), and (ii) standardizes outputs for instruction tuning and RL.

**Text Grammar.** We constrain outputs with a minimal JSON-like grammar to reduce decoding entropy, and they can be parsed automatically at inference:

$$\underbrace{\texttt{[[x, y], [x, y], ...]}}_{\text{polygon}}, \underbrace{\texttt{[x, y]}}_{\text{point}}, \underbrace{\texttt{[x1, y1, x2, y2]}}_{\text{bbox}}.$$

**Data Annotation Pipeline.** To scale our framework with large-scale web data, we construct an automatic data annotation pipeline, as shown in Fig. 3, to generate instance-level segmentation labels.

Specifically, the pipeline employs: Grounding-DINO (Liu et al., 2023) for phrase grounding and object detection, SAM to extract segmentation masks, the algorithm for converting mask to contour coordinates, and an off-the-shelf VLM for optional, refined object description tagging.

## 3.2 Training Pipeline of SimpleSeg

Our training including: (i) instruction tuning (SFT) to cold-start structured generation, and (ii) reinforcement learning (RL) to optimize sequence-level, location-aware objectives.

**Stage I: Instruction Tuning.**    According to the aforementioned polygon-based representation of masks, we curate instruction–response pairs spanning (text↔point), (text↔bbox), and (text/point→mask). The supervised finetuning stage aims to teach the MLLM to emit correct output formats, including well-formed coordinates, closing brackets, and consistent ordering, while learning basic grounding priors. Empirically, this already yields competitive performance and provides a stable initialization for RL.

**Stage II: Reinforcement Learning with GSPO.**    Reinforcement learning (RL) has demonstrated significant effectiveness in reasoning tasks for MLLM (Shao et al., 2024; Guo et al., 2025; Team et al., 2025a). However, the potential of RL for sharpening fine-grained perception remains largely untapped. While SFT aligns tokens to local supervision, pixel-level segmentation quality depends on global properties of the entire sequence (closure, boundary fidelity, and verbosity). We focus on leveraging RL to boost the perception accuracy of MLLM, since, in essence, reinforcement learning is a more reasonable and efficient optimization method for perception tasks. Especially under our data and task formulation, we are not aiming to force the model to rigidly regress fixed ground-truth coordinates in the training data, as contour sequences are inherently flexible. The optimization process relies more on location-aware rewards to explore and refine predictions at the sequence level, while format-rule-based judges can enforce valid, parseable output structures. We therefore adopt GSPO (Zheng et al., 2025) as our RL algorithm, and adopt a rule-based reward system that mainly consists of three types of rewards:

- **Mask IoU** reward: The direct IoU between the predicted and ground-truth mask. The range of reward values is $[0.0, 0.1]$. We set a threshold $\tau$ that the reward is 0 if IoU is less than $\tau$.
- **MSE Distance IoU** reward: The negative mean square distance between the centroids of predicted and ground-truth mask. It is normalized with the image size.
- **Format** reward: In addition to the accuracy reward model, we employ a format reward that enforces the model to output correct polygon coordinates formats. If the format is wrong, the reward returns zero.

Crucially, RL lets the model discover alternative yet valid trajectories (e.g., different starting points, equivalent vertex sets) instead of overfitting to a single annotation.

**Why RL for Point Trajectories?**    As far as we know, we are the first to leverage RL in the realm of decoder-free MLLM segmentation. Contours are inherently many-to-one w.r.t. masks; enforcing exact token matching is suboptimal. Reinforcement learning well bridges the gap and evaluates the *rendered mask*, directly aligning optimization with the end metric, and improves closure and thin-structure adherence that are difficult to teach via token-level losses alone.

## 4 Experiment

### 4.1 Implementation Details

Our method builds on the open-source MLLM *Kimi-VL* (Team et al., 2025b), an efficient MoE model with 2.8B activated parameters. Training uses 32 NVIDIA GPUs with a global batch size of 256 and the enhanced Muon optimizer (Liu et al., 2025a). For supervised fine-tuning (SFT), we use an initial learning rate of $5 \times 10^{-5}$ with cosine decay to $2 \times 10^{-6}$, and a warm-up ratio of 0.03. For reinforcement learning (RL), we adopt GSPO with clip ratio in $[3 \times 10^{-4}, 4 \times 10^{-4}]$ and a KL coefficient of 0.01. Unless otherwise specified, coordinates are normalized and serialized in the text space using our polygon format, and they are sparsified by a tolerance parameter $\epsilon$ (cf. Sec. 4.3).

Table 1: **Referring Expression Segmentation** results (cIoU) on refCOCO (+/g) datasets (Kazemzadeh et al., 2014; Mao et al., 2016), compared to approaches that adopt MLLMs for segmentation. The MLLM instance of Text4Seg in the table is InternVL2-8B.

| Methods | refCOCO | | | refCOCO+ | | | refCOCOg | | Avg. |
|---|---|---|---|---|---|---|---|---|---|
| | val | testA | testB | val | testA | testB | val | test | |
| *Decoder-based Models* | | | | | | | | | |
| NEXT-Chat (Zhang et al., 2023) | 74.7 | 78.9 | 69.5 | 65.1 | 71.9 | 56.7 | 67.0 | 67.0 | 68.9 |
| LISA (Lai et al., 2024) | 74.9 | 79.1 | 72.3 | 65.1 | 70.8 | 58.1 | 67.9 | 70.6 | 69.9 |
| PixelLM (Ren et al., 2024) | 73.0 | 76.5 | 68.2 | 66.3 | 71.7 | 58.3 | 69.3 | 70.5 | 69.2 |
| AnyRef (He et al., 2024) | 76.9 | 79.9 | 74.2 | 70.3 | 73.5 | 61.8 | 70.0 | 70.7 | 72.2 |
| GSVA (Xia et al., 2024) | 77.2 | 78.9 | 73.5 | 65.9 | 69.6 | 59.8 | 72.7 | 73.3 | 71.4 |
| LaSagnA (Wei et al., 2024) | 76.8 | 78.7 | 73.8 | 66.4 | 70.6 | 60.1 | 70.6 | 71.9 | 71.1 |
| Groundhog (Zhang et al., 2024b) | 78.5 | 79.9 | 75.7 | 70.5 | 75.0 | 64.9 | 74.1 | 74.6 | 74.2 |
| Text4Seg (w/ SAM) | 79.2 | 81.7 | 75.6 | 72.8 | 77.9 | 66.5 | 74.0 | 75.3 | 75.4 |
| *Decoder-free Models* | | | | | | | | | |
| Text4Seg (Lan et al., 2024) | 74.7 | 77.4 | 71.6 | 68.5 | 73.6 | 62.9 | 70.7 | 71.6 | 71.4 |
| **SimpleSeg** | 76.9 | 78.9 | 73.6 | 71.1 | 75.2 | 66.1 | 72.8 | 74.3 | 73.6 |

Table 2: **Referring Expression Comprehension** results (Acc@0.5) on RefCOCO (+/g) datasets, compared to approaches that adopt MLLMs for segmentation.

| Methods | refCOCO | | | refCOCO+ | | | refCOCOg | | Avg. |
|---|---|---|---|---|---|---|---|---|---|
| | val | testA | testB | val | testA | testB | val | test | |
| *Decoder-based Models* | | | | | | | | | |
| LISA (Lai et al., 2024) | 85.4 | 88.8 | 82.6 | 74.2 | 79.5 | 68.4 | 79.3 | 80.4 | 79.8 |
| GSVA (Xia et al., 2024) | 86.3 | 89.2 | 83.8 | 72.8 | 78.8 | 68.0 | 81.6 | 81.8 | 80.3 |
| NEXT-Chat (Zhang et al., 2023) | 85.5 | 90.0 | 77.9 | 77.2 | 84.5 | 68.0 | 80.1 | 79.8 | 80.4 |
| PixelLM (Ren et al., 2024) | 89.8 | 92.2 | 86.4 | 83.2 | 87.0 | 78.9 | 84.6 | 86.0 | 86.0 |
| Text4Seg (w/ SAM) | 90.3 | 93.4 | 87.5 | 85.2 | 89.9 | 79.5 | 85.4 | 85.4 | 87.1 |
| *Decoder-free Models* | | | | | | | | | |
| Text4Seg (Lan et al., 2024) | 88.3 | 91.4 | 85.8 | 83.5 | 88.2 | 77.9 | 82.4 | 82.5 | 85.0 |
| **SimpleSeg** | 90.5 | 92.9 | 86.8 | 85.3 | 89.5 | 80.2 | 86.1 | 86.5 | 87.2 |

## 4.2 MAIN RESULTS

**Referring Expression Segmentation**    The referring expression segmentation (RES) task aims to segment the object in an image that is described by a given natural-language expression. We follow the training recipe of (Lan et al., 2024), which constructs the training dataset with the `train` split of refCOCO, refCOCO+ (Kazemzadeh et al., 2014), refCOCOg (Mao et al., 2016), and refCLEF. As shown in Tab. 1, our SimpleSeg achieves superior performance in decoder-free models, and comparable to decoder-based methods. This demonstrates our method's strong fine-grained perception capacity as a generalist vision-language model, without any modification of model architecture.

**Referring Expression Comprehension**    Our SimpleSeg is also directly usable for object detection by converting the predicted mask to a bounding box through simple min-max operations. Accordingly, we evaluate our approach on the Referring Expression Comprehension (REC) task, using the same model trained as in RES. For the evaluation specification, we calculate the average accuracy with a threshold IoU of 0.5 between the predicted and ground truth bounding boxes. As shown in Tab. 2, our SimpleSeg obtains state-of-the-art performance on the benchmarks. Specifically, our method achieves an average score of 87.2, exceeding the closest competitor, Text4Seg, even though it is equipped with a mask refiner.

> **Takeaway3**  While still not perfect, SimpleSeg demonstrates that a minimalist, decoder-free MLLM achieves performance on challenging segmentation benchmarks that is comparable to complex models augmented with specialized decoders.

Table 3: The gIoU score with different training stages in validation sets.

| Pre-train | SFT | RL | refCOCO | refCOCO+ | refCOCOg |
|:---:|:---:|:---:|:---:|:---:|:---:|
| | ✓ | | 65.5 | 60.8 | 60.4 |
| | ✓ | ✓ | 75.2 (↑ 9.7) | **70.6** (↑ 9.8) | 70.9 (↑ 10.5) |
| ✓ | | | 25.3 (↓ -45.7) | 18.7 (↓ -46.4) | 25.7 (↓ -43.0) |
| ✓ | ✓ | | 70.1 (↑ 4.6) | 65.0 (↑ 4.2) | 65.7 (↑ 5.3) |
| ✓ | ✓ | ✓ | **78.5** (↑ 13.0) | 69.8 (↑ 9.0) | **71.7** (↑ 11.3) |

## 4.3 EXPLORATION STUDIES

**Effect of Training Stages** Table 3 ablates on the effect of different training stages, including pre-training, SFT, and RL. We evaluate the gIoU score on the validation set of different datasets. SFT alone reaches **65.5**, **60.8**, and **60.4** gIoU on three datasets, establishing a strong baseline from purely supervised polygon learning. Adding RL lifts performance to **75.2** (+9.7), **70.6** (+9.8), and **70.9** (+10.5) respectively by a large margin, indicating that sequence-level credit assignment with IoU-based rewards is important for accurate *closed* polygon generation and token-economical outputs. Pre-training without SFT performs poorly (25.3 gIoU), and this stems from a distribution shift between pre-training and SFT prompts. Pre-training lacks RefCOCO-style questions, and this phase primarily focuses on utilizing weakly labeled data to establish the

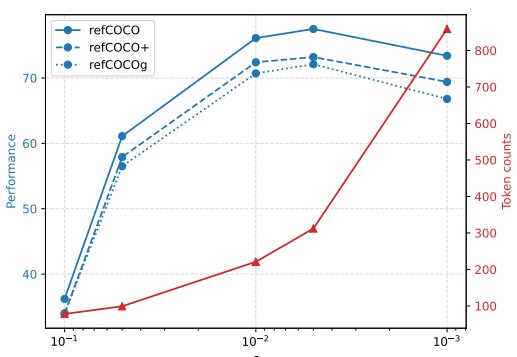

Figure 4: The relationship between the sequence length and performance under the control of the point density parameter $\epsilon$.

model's basic segmentation ability. It can be seen that both SFT and SFT+RL benefit significantly from pre-training, respectively, raising the gIoU by **4.6** and **13.0** on refCOCO, confirming that scaling the training data strengthens perceptual priors and benefits downstream tasks.

> **Takeaway4** A sweet spot — between the model's capacity for sequential understanding and the contour's geometric fidelity is crucial for achieving effective geometric reasoning. And Reinforcement Learning automatically finds it.

**Point/Polygon Density** ($\epsilon$)   $\epsilon$ is a hyperparameter that controls the polygon approximation accuracy for mask-to-contour conversion. A smaller $\epsilon$ yields higher polygon precision and thus more sampled points. Fig 4 varies the sparsification tolerance. We conduct SFT experiments with different $\epsilon$ and the results are demonstrated in Fig. 4. Performance is *unimodal* w.r.t. token length: too few points underfit shapes (35.6 cIoU at 78 tokens), too many induce long-horizon decoding errors and length exposure (72.5 cIoU at 859 tokens), while a moderate density (221 tokens) yields the best score.

Figure 5: gIoU score with different rewards.

| Reward | RefC | RefC+ | RefCg |
|:---:|:---:|:---:|:---:|
| IoU | 76.9 | 71.9 | 72.9 |
| + Distance | 77.1 | 72.2 | 73.1 |
| + Length Penalty | 66.7 | 62.4 | 62.0 |

**Reward Design for RL** Table 5 studies the effectiveness of different reward components, including distance and length penalty. We observe that the involvement of distance yields an average gain of around 0.2, while imposing hard length constraints degrades the performance.

**Metrics Trends During RL** In Fig. 6, we illustrate the training states during the reinforcement learning process, including the reward, response length, and validation performance, on different settings of $\epsilon$. One observation is that, even without length-related rewards, the model can adaptively adjust its output length to keep a reasonable accuracy-efficiency balance during RL. With a large density of $\epsilon = 0.001$, the number of tokens decreases moderately, trading redundant vertices for

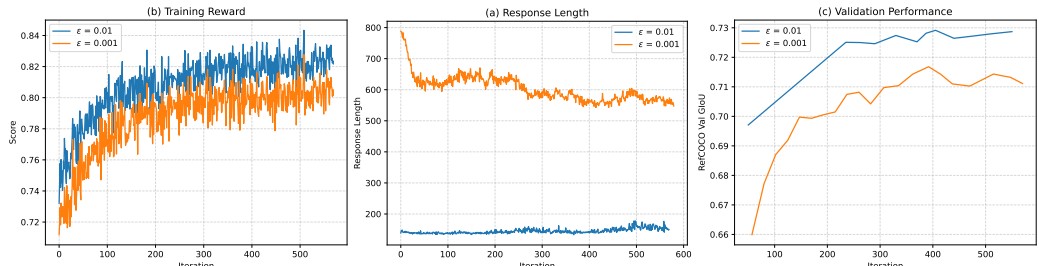

Figure 6: The curve of metrics during the RL stage.

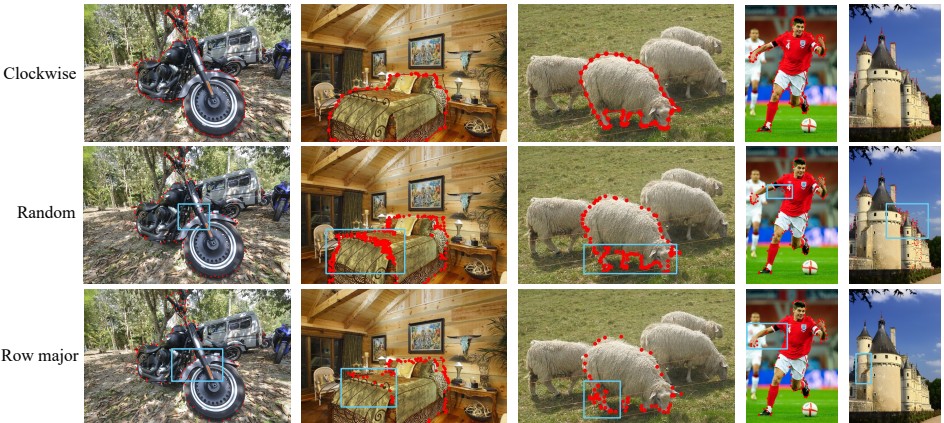

Figure 7: Visual results with different training orders of sampling points. Unsatisfactory prediction points are marked with blue boxes in the image.

token efficiency while preserving mask fidelity. When the token budget is low with $\epsilon = 0.01$, the response length is increased slightly to better refine the segmentation results.

**Order of Sampling Points** To convert a binary mask to contour coordinates, we apply the Suzuki-Abe algorithm for boundary tracing, which can keep the sampling points in clockwise order. We conduct experiments with different organizations of sampling points, and display the results in Fig. 7. First, without enforcing clockwise ordering, the coordinate sequence fails to form a valid polygon, and no segmentation mask can be derived. Second, clockwise ordering provides a more principled learning target, reducing model entropy. As shown in the figure, alternative orders confuse the model and yield chaotic or repeated points, decreasing the token efficiency.

**Results on extended tasks** As mentioned in Sec. 3.1, we extend the perception task via a unified query interface beyond just the query of the reference phrase. Namely, our SimpleSeg can also achieve the SAM-like functionality, such as `point → mask` and `bbox → mask`. We visualize the results in the Appendix due to limited space. This significantly demonstrates the generality of our framework, further pushing the upper limit of MLLMs' versatile perception capacity.

## 5 CONCLUSION

In this work, we demonstrated that a strikingly simple approach—reframing segmentation as the prediction of a sequence of points—is sufficient to unlock a powerful, native capability for pixel-level perception latent within standard MLLM architectures. Our model, SimpleSeg, cultivated through a novel SFT→RL pipeline, achieves performance that is comparable to, and often surpasses, complex decoder-based systems. It is a finding that high-fidelity perception can be an emergent property of MLLMs. Our work paves the way for a new generation of truly generalist multimodal systems that seamlessly unify perception and reasoning within a single, elegant framework.

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

## A    LIMITATIONS AND DIAGNOSTICS

While SimpleSeg eliminates task decoders, long sequences remain a bottleneck for high-resolution, highly-curved objects. Errors tend to cluster at sharp corners and thin structures under aggressive sparsification. Future diagnostics should consider boundary F-score, vertex-wise Chamfer distance, and token-per-mask analyses across object scales to complement cIoU/Acc@0.5.

## B    ADDITIONAL IMPLEMENTATION DETAILS

Tab. 4 and Tab. 5 present the training hyperparameters used in SFT and RL stages.

Table 4: Hyper-parameters and training settings for SFT stage.

|  | Param Name | Value |
|---|---|---|
| Optimizer | Type | Enhanced Muon |
|  | Max Learning rate | 5e-5 |
|  | Max Learning rate | 2e-6 |
|  | Weight decay | 0.1 |
|  | $(\beta_1, \beta_2)$ | (0.9, 0.95) |
|  | Gradient norm clip | 1.0 |
|  | Scheduler | Cosine decay |
|  | Warmup ratio | 0.03 |
| Training | Numerical precision | FP16 |
|  | Global batch size | 256 |
|  | Number of samples per epoch | 800k |
|  | Total epochs | 1 |

Table 5: Hyper-parameters and training settings for RL stage.

|  | Param Name | Value |
|---|---|---|
| Algorithm | Type | GSPO |
|  | Clip Ratio | [3e-4, 4e-4] |
|  | KL Alpha | 0.1 |
|  | Num. Responses per Group | 8 |
| Optimizer | Type | Enhanced Muon |
|  | Constant Learning rate | 2e-6 |
|  | $(\beta_1, \beta_2)$ | (0.9, 0.95) |
|  | Gradient norm clip | 1.0 |
| Training | Numerical precision | FP16 |
|  | Rollout Temperature | 0.8 |
|  | Global batch size | 256 |
|  | Number of samples per epoch | 800k |
|  | Total epochs | 2 |

## C    ADDITIONAL QUALITATIVE RESULTS

As shown in Fig. 8, Fig. 9, and Fig. 10, we provide more example results of SimpleSeg on our extended tasks, which significantly demonstrate our SimpleSeg's accuracy, robustness, and generalization.

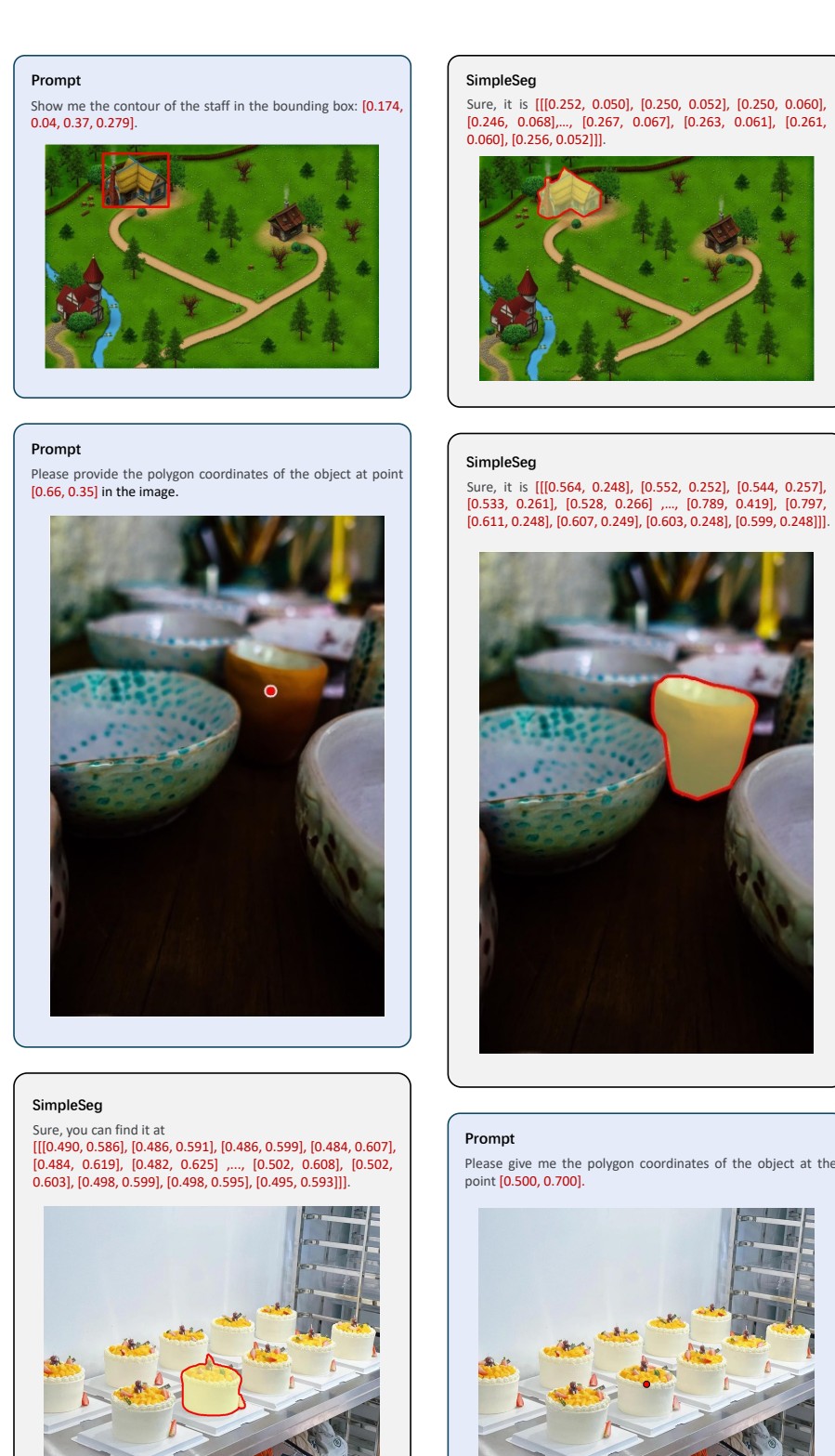

Figure 8: More results on more diverse tasks, including (point→mask) and (bbox→mask). The position information is visualized in the image.

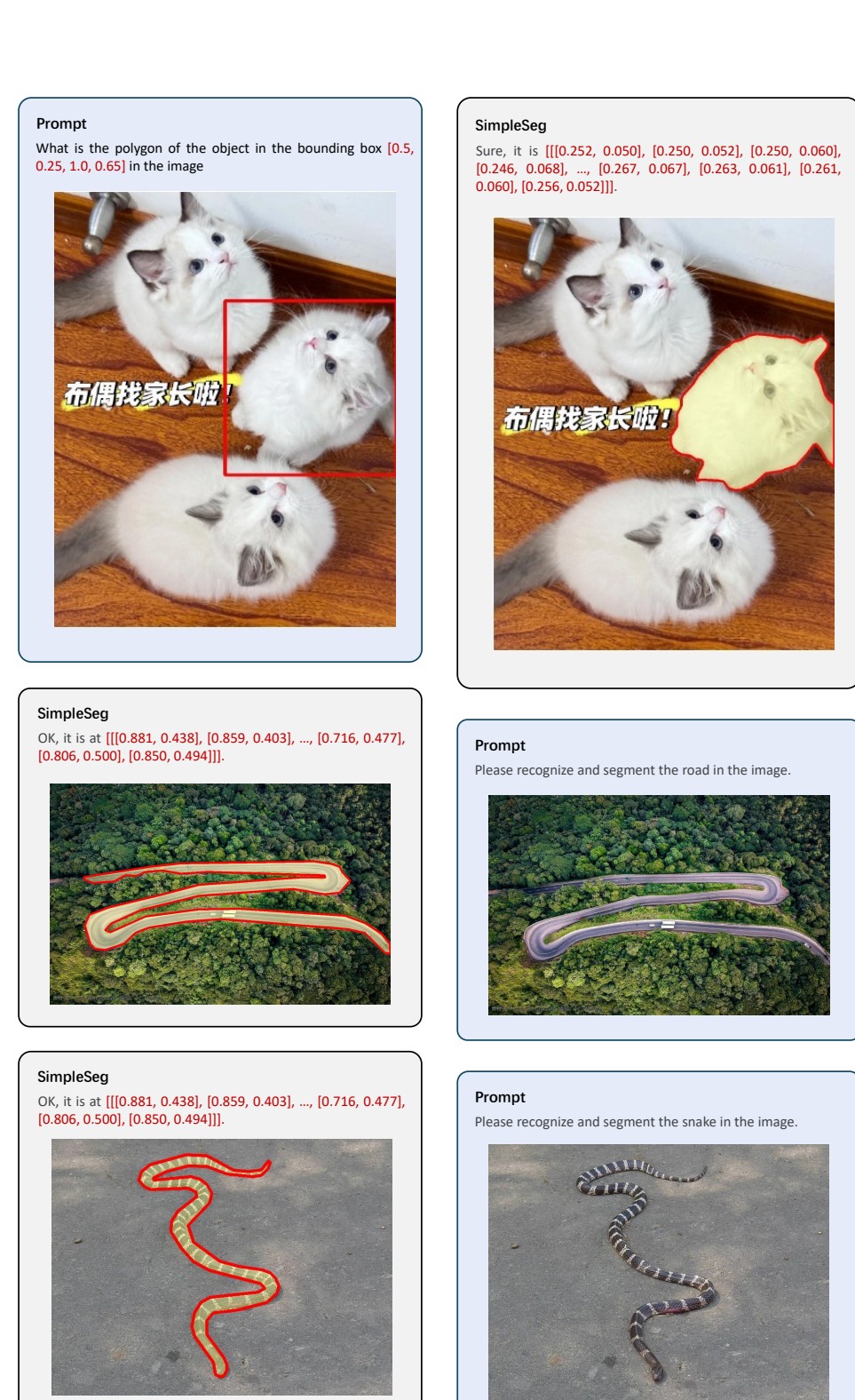

Figure 9: More results on more diverse tasks including (bbox→mask) and (text→mask). The position information is visualized in the image.

**Prompt**

Please point out the middle giraffe in the image.

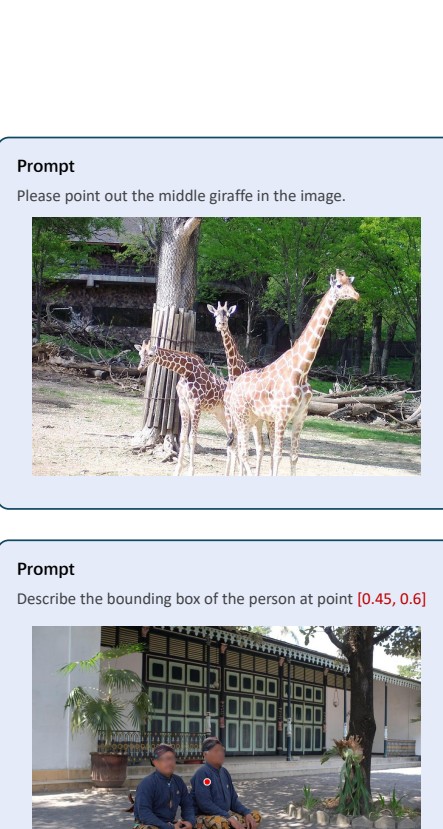

**SimpleSeg**

It is at [0.483, 0.396].

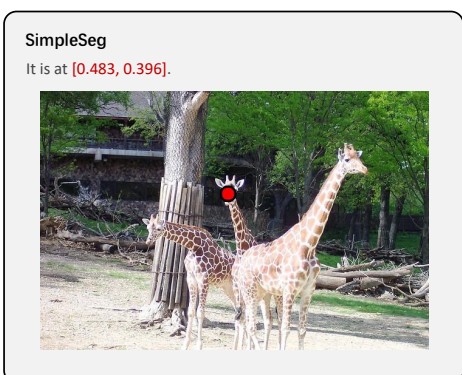

**Prompt**

Describe the bounding box of the person at point [0.45, 0.6]

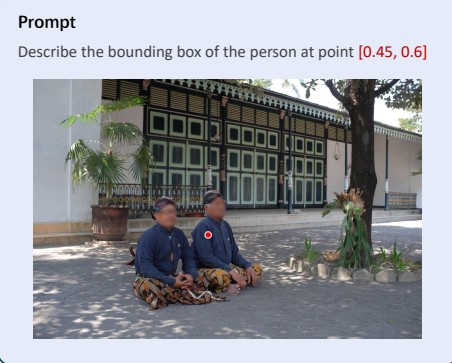

**SimpleSeg**

Sure, the bounding box is [0.404, 0.423, 0.590, 0.832].

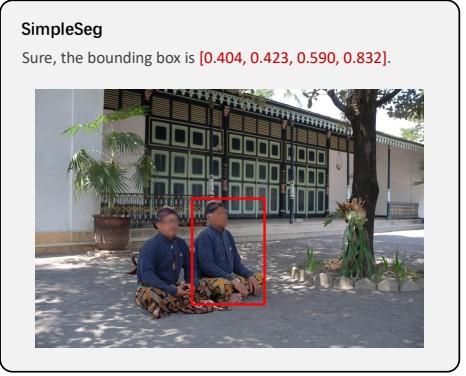

**Prompt**

Bounding box of the dog on the seat.

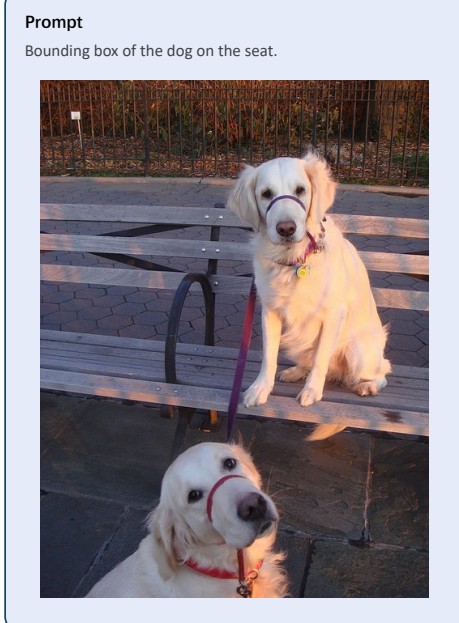

**SimpleSeg**

The bounding box is [0.475, 0.127, 0.906, 0.633].

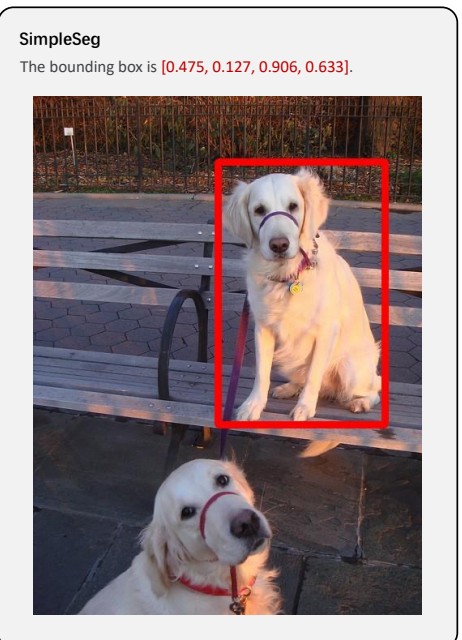

Figure 10: More results on more diverse tasks, such as (text→point) and (text→bbox). The position information is visualized in the image.

