# Towards Pixel-level MLLM Perception via Simple Points Prediction Rebuttal Supplementary Materials

## 1 More Results

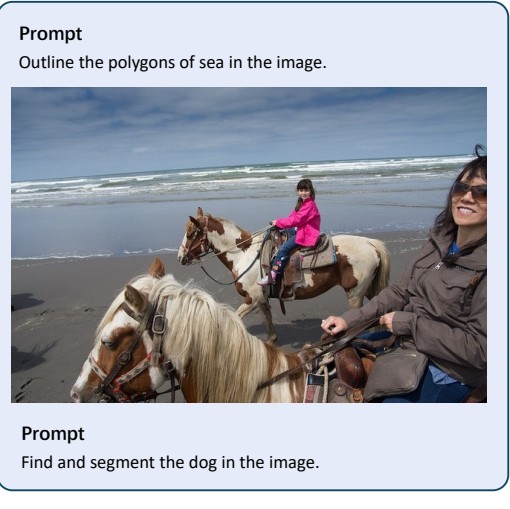

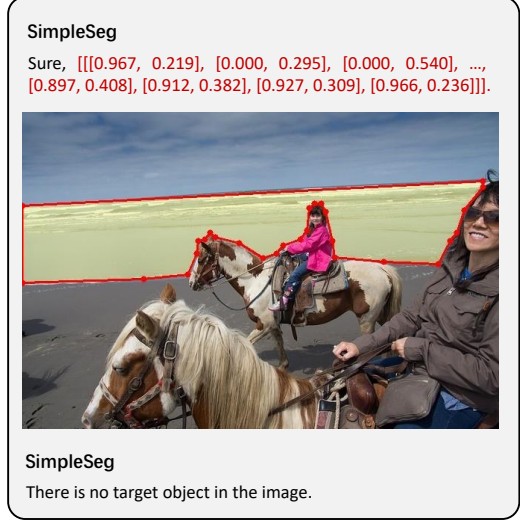

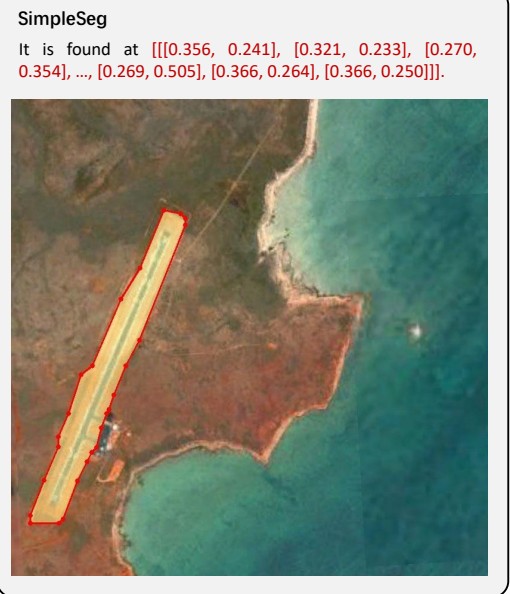

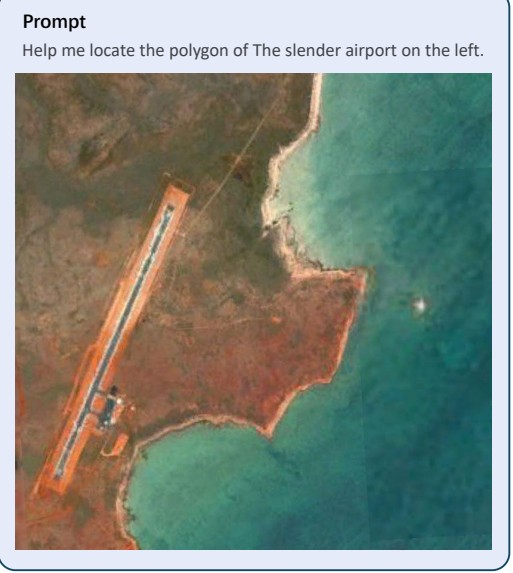

Figure 1: More results of background stuff segmentation, non-existent objects, and remote sensing image segmentation.

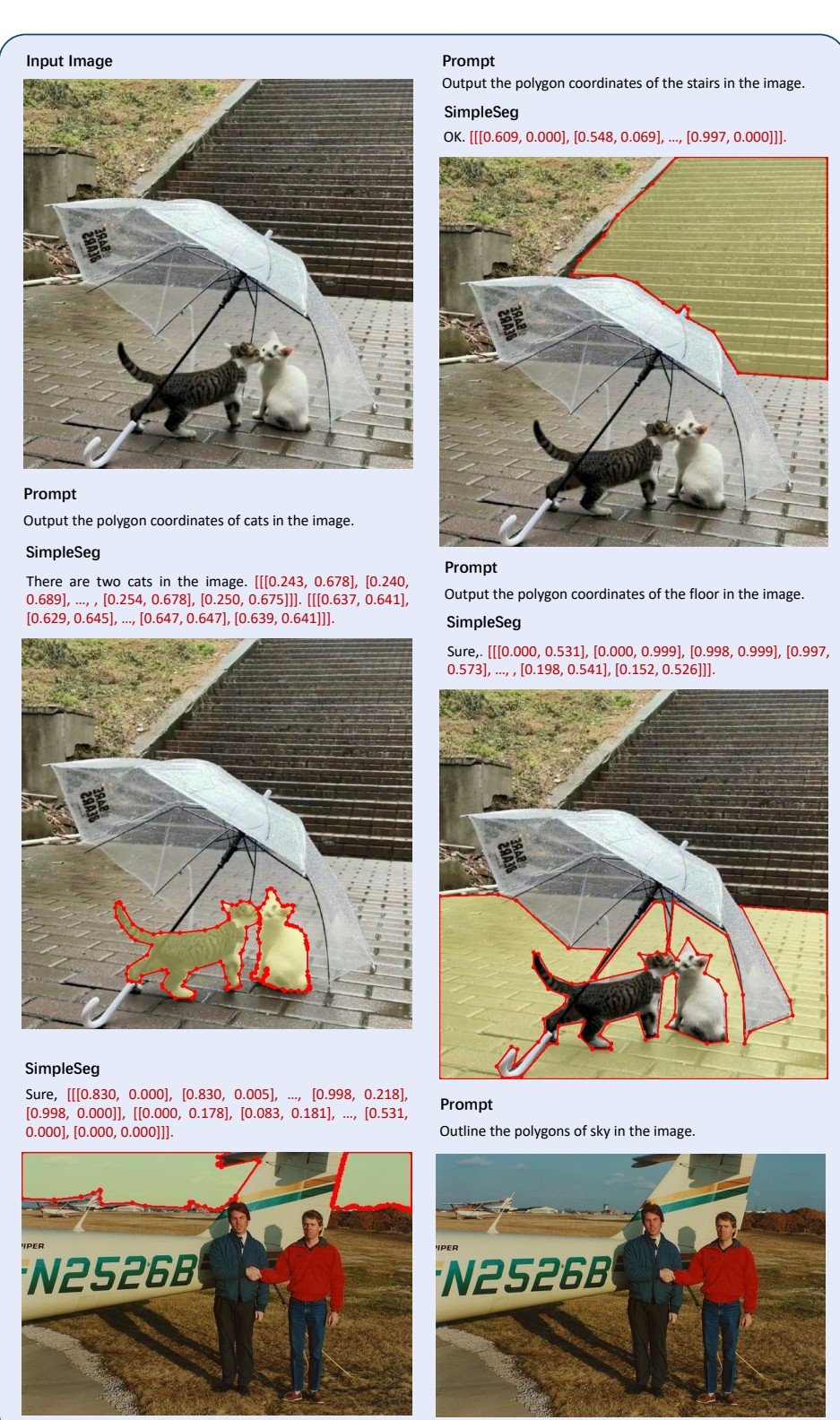

Figure 2: More results of panoptic segmentation, multiple objects segmentation, and multi-parts object segmentation.

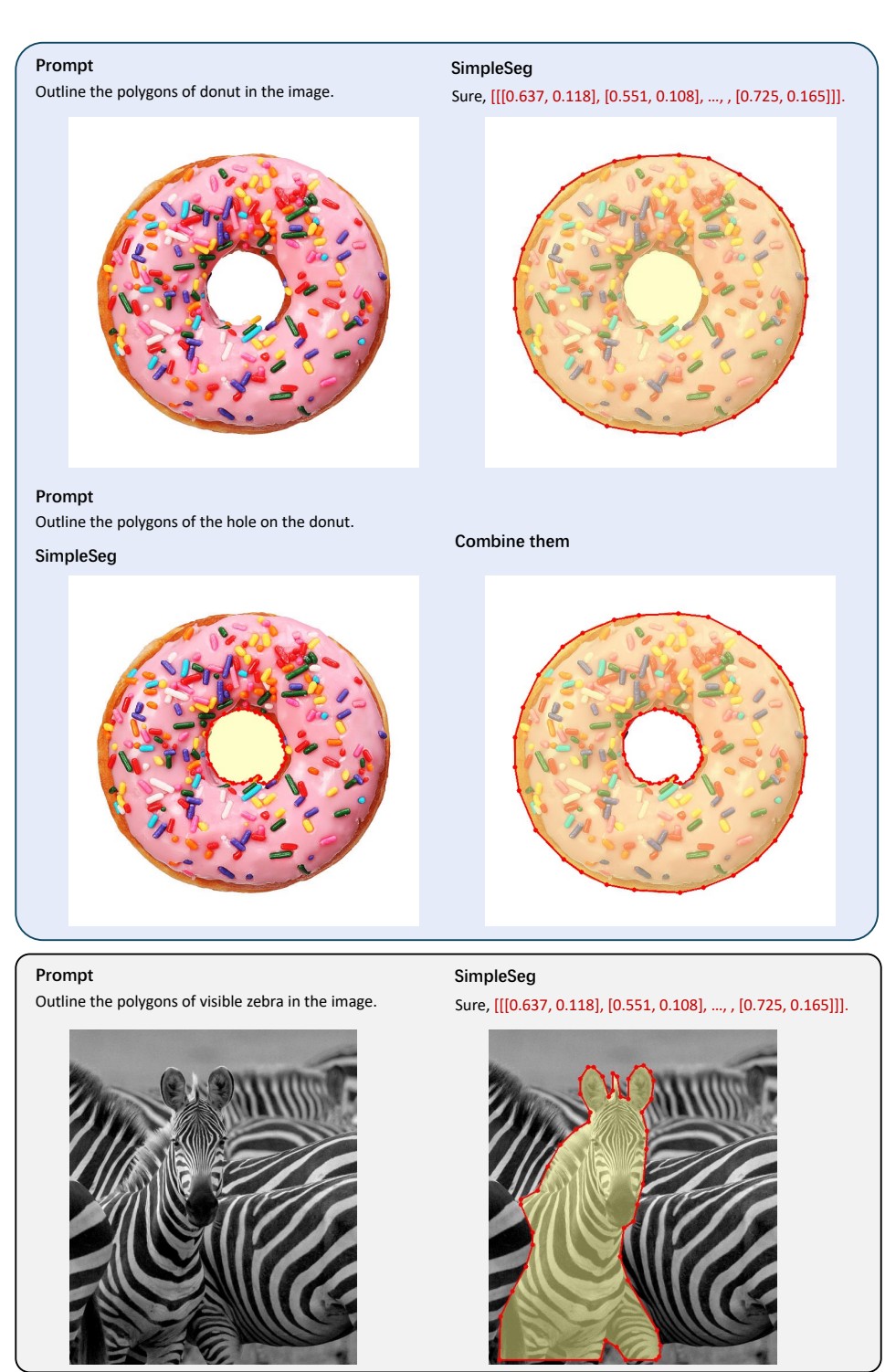

Figure 3: Visual results of failure cases, e.g., object with holes and texture confusion.