# OpenReview forum: "Towards Pixel-level VLM Perception via Simple Points Prediction"
_ICLR.cc/2026/Conference — Submitted to ICLR 2026_

### Official Review · Reviewer_nBrQ · 2025-10-20

**Soundness:** 1
**Presentation:** 2
**Contribution:** 1
**Rating:** 2
**Confidence:** 5

**Summary:**

The paper proposes SimpleSeg, which models segmentation as predicting the vertex coordinates of object boundaries, thereby avoiding the use of complex segmentation decoders. The paper also trains the model using reinforcement learning (RL) with an IoU-based reward to improve performance.

**Strengths:**

1. Exploring how to perform segmentation using the MLLM's inherent capabilities, rather than relying on a segmentation decoder, is a meaningful research direction.
2. Predicting text-based coordinates allows for a unified training objective with the MLLM's other tasks.

**Weaknesses:**

1. Using polygons to model segmentation introduces approximation errors and limits expressive power. For example, in Figure 2, a polygon with a finite number of vertices cannot perfectly fit the line graph and the lightning bolt. Furthermore, the paper only considers cases with a single object boundary, whereas in real-world scenarios, many objects and backgrounds (e.g., a donut) cannot be represented by a single polygon.
2. Predicting text-based coordinates requires outputting a large number of text tokens, which significantly increases inference time.
3. The paper lacks comparisons with some recent works, including VistaLLM[1], HiMTok[2], UFO[3], and SAM4MLLM[4]. Notably, neither VistaLLM nor UFO requires a segmentation decoder, and VistaLLM also predicts polygon vertices.
4. The paper's segmentation performance shows a significant gap compared to state-of-the-art decoder-based methods.

[1] Jack of all tasks master of many: Designing general-purpose coarse-to-fine vision-language model. CVPR 2024

[2] HiMTok: Learning Hierarchical Mask Tokens for Image Segmentation with Large Multimodal Model. ICCV 2025

[3] UFO: A Unified Approach to Fine-grained VisualPerception via Open-ended Language Interface. NeurIPS 2025

[4] SAM4MLLM: Enhance Multi-Modal Large Language Model for Referring Expression Segmentation. ECCV 2024

**Questions:**

1. The method is only evaluated on Kimi-VL. Can it be extended to QwenVL or InternVL?
2. After the SFT and RL training described in the paper, what are the changes to the model's general VQA performance?
3. Does the model support multi-object segmentation?

---

> ### Author Response · Authors · 2025-11-26
>
> Thanks for your valuable feedback on our manuscript.
>
> **Q.1 Concerns about the limitation of the expressive power of polygon representation.**
>
> We appreciate the reviewer's insightful comments regarding the potential limitations of polygon representation.
>
> - **Minimizing Approximation Errors:** We acknowledge that any finite-vertex polygon introduces approximation errors. However, our proposed method addresses this limitation by not strictly relying on a fixed-vertex polygon representation. Instead, our approach models the object boundary as a contour representation where we significantly scale up the number of predicted points. By increasing the density of the predicted points along the boundary (i.e., scaling up the number of vertices), the representation approaches a high-fidelity curve/contour, which substantially reduces the approximation error.
>
> - **Handling Real-World Complex Objects:** We want to clarify that our model is not limited to considering a single object boundary. Our method is capable of handling complex, real-world segmentation scenarios, including **Multiple Disjoint Objects**, **Objects with Disconnected Parts**, and **Background Segmentation**. Objects with holes like donuts can be processed correctly by merging two segmentation results. We have updated the supplementary material (PDF) to include an expanded set of visual results that clearly demonstrate our model's capability in all of these scenarios.
>
> - **Core Targets:** We agree with the perspective that polygon approximation, if used as the final output format, has limitations. However, Segmentation is not the core target of this paper; it is a means to an end. Our primary objective is to explore and demonstrate the Pixel-Level Perception Capability of Vision-Language Models (VLMs). We utilize the task of pixel-level localization (segmentation/contour prediction) as a rigorous and quantifiable proxy to provide evidence that VLMs inherently possess, or can be effectively endowed with, the ability to perform fine-grained, pixel-accurate perception, which goes significantly beyond coarse bounding box or global image understanding. Our main contribution lies in unveiling and harnessing this pixel-level capacity within VLMs. The successful generation of high-fidelity boundaries/contours serves to prove the existence of this enhanced perception, rather than offering a new state-of-the-art segmentation algorithm itself.
>
> - **Generalization to Unconventional Shapes**: We emphasize that the examples shown in Figure 2, such as the line graph charts and cartoon graphics, represent a significant advancement. To the best of our knowledge, most prior segmentation works focus on realistic image datasets like RefCOCO, which do not contain such highly stylized or data-intensive shapes. Crucially, these challenging cases were not present in our training data. Our successful segmentation of these shapes is an emergent capability of our VLM-based approach, demonstrating superior generalization power. This success highlights how we effectively leverage the VLM's powerful general-purpose knowledge and reasoning, achieving capabilities that are typically absent in traditional segmentation models.
>
>
> **Q.2 Inference Time**
>
> We appreciate the reviewer raising the trade-off regarding inference time. However, this concern overlooks the strategic objective and unique capability that our method introduces to the field of VLMs.
>   - The VLM's Enhanced Perception is the Priority. Our work is fundamentally a VLM enhancement paper, not merely a pursuit of incremental speed gains in a dedicated segmentation task. The primary purpose of our design is to establish a robust, text-based mechanism for spatial grounding and fine-grained visual perception within the VLM's native generative architecture.
>   - Segmentation is the Enabler, Not the Bottleneck. We want to be clear: Segmentation is the means to validate the VLM's enhanced spatial grounding, not the ultimate end-goal of our entire framework. Allow the VLM to perform complex downstream reasoning that requires precise spatial grounding. (e.g., "Find the line graph, tell me its equation, and then outline the area above the line"). The increased inference time is a necessary trade-off for unlocking these powerful emergent perception and grounding abilities within the VLM framework, which is currently beyond the reach of standard VLMs.

---

> ### Author Response · Authors · 2025-11-26
>
> **Q.3 Discussion about more recent works and performances**
>
> Thank you for raising the concern regarding comparisons with several recent and relevant works. We agree that these methods are important and should be considered, and we will cite and integrate comparisons with these methods into the revised manuscript.
>
> While methods like VistaLLM and UFO demonstrate impressive segmentation and localization capabilities without a dedicated segmentation decoder, their core contribution often lies in efficient tokenization or the integration of segmentation as a downstream task on top of a Vision Language Model (VLM). Our work focus on prove that the general VLM inherently possesses or can acquire powerful, high-resolution, pixel-level perceptual capabilities when the segmentation information (contour) is properly encoded and scaled within the LLM's token space.
>
> Compared to VistaLLM, we scale up the number of vertices and data sources, which demonstrate superior performance and generalizability.
> **As mentioned above, we aim to prove the potential fine-grained perception of VLM, rather than offering a new state-of-the-art segmentation algorithm itself**. The success of our method demonstrates that the VLM is not just an interface layer but can effectively perform fine-grained pixel-level regression when given the right input/output structure. This is a critical step towards realizing the ultimate goal of general-purpose VLM agents that can handle both high-level reasoning and low-level perception, which goes beyond simply achieving state-of-the-art on a few benchmarks.

---

### Official Review · Reviewer_PQGm · 2025-10-22

**Soundness:** 3
**Presentation:** 2
**Contribution:** 3
**Rating:** 2
**Confidence:** 4

**Summary:**

The paper enhances the capability of MLLM in the pixel-level prediction task via task reformulation and reinforcement learning. It modules the localization as a 4-tuple [text, point, box, mask], where mask is represented by polygon with a consistennt clockwise ordering. The RL algorithm enhances the learning flexibility of this mask representation.

**Strengths:**

It is novel that the paper incorporates reinforcement learning into pixel-level prediction of MLLM.

**Weaknesses:**

1. The mask representation of a polygon with the clock-wise vertex order is not new, which has been done in PolyFormer [ref1].

2. The data construction is not clear enough. Lack of implementation details, which reduces the reproducibility. What is the total number of training images? What is the data composition of SFT and RL training? What is the source data? Simply calling web data is not acceptable. What are the model versions of Grounding-DINO, SAM, and VLM? And there is no open-source statement for the data construction pipeline.

3. What is the sequence of the three training stages? What is the training/inference time cost and the GPU model. What is the exact meaning of Pre-train in Table 3?

4. Page 6 line 312, what is the meaning of ''thin-structure adherence''? Page 8 line 418, it cannot be seen from Fig. 4 that the performance is 53.7 cIoU at 78 tokens. It is below 40.

Considering the above questions and missing parts, I hold a negative rating towards this paper. I might change my mind if these questions are addressed well.

[ref1] Liu J, Ding H, Cai Z, et al. Polyformer: Referring image segmentation as sequential polygon generation. CVPR 2023: 18653-18663.

**Questions:**

see weaknesses

---

> ### Author Response · Authors · 2025-11-25
>
> Thank you for taking the time to provide your valuable feedback. Your insights are extremely helpful and have provided us with clear directions for improvement.
>
> **Q.1 Concerns about a lack of Novelty**
>
> We acknowledge that the use of a polygon representation with a clock-wise vertex order is not a novel technique, as correctly pointed out by the reviewer with the reference to PolyFormer. While the sequential vertex-ordering formulation shares a conceptual resemblance, we wish to emphasize several critical and fundamental differences that distinguish our approach and highlight its novelty.
>
> - First, PolyFormer is  specialized Encoder-Decoder structure (similar to ALBEF) designed primarily for segmentation. The polygon representation is the end product of a dedicated segmentation model. Our work has different targets and is based on a general VLM, and the contour representation is designed to be fully compatible with and generatable by the LLM backbone, enabling a unified text-only representation across all tasks, which PolyFormer cannot achieve.
> - Second, PolyFormer's approach is tailored specifically for the segmentation task, leading to dedicated segmentation losses. In contrast, we utilize a pure text-based representation for the mask, allowing us to leverage the standard NTP training objectives during SFT stage. This unified formulation is a key enabler for the scalability and generalization of VLM. Besides, we innovatively explore RL to leverage sequence-level supervision of task-specific metrics.
> - Third, we significantly scale up the number of vertices to represent a dense contour. This change moves the representation from a coarse polygon (suitable for simplified tasks) to a high-fidelity contour (crucial for detailed segmentation and higher resolution).
> - Fourth, we have conducted extensive experiments involving significant data scaling and demonstrated the model's robustness and generalization capability on out-of-distribution data, shown in Figure 2 in the paper, such as the line graph charts and cartoon graphics, represent a significant advancement, which PolyFormer cannot achieve.
>
> The core target goal of our work is, by achieving strong performance across dense perception task like segmentation, we prove that the large multimodal model inherently possesses/can possess pixel-level perception, rather than merely developing a new segmentation technique or "chasing benchmark scores."
>
> **Q.2 Implementation Details of Data Construction**
>
> We apologize for any lack of clarity regarding the data composition, and we want to rectify this confusion immediately.
>
> - First, the SFT training data composition is fully aligned with that of Text4Seg [1].
>     - All the main results reported in the paper (e.g., Table 1, Table 2, comparing against SOTA) are obtained using a model trained exclusively on the RefCOCO dataset (RefCOCO/RefCOCO+/RefCOCOg/RefClef) for a fair comparison with existing methods.
>     - Both the Supervised Fine-Tuning (SFT) and Reinforcement Learning (RL) stages were trained only on the RefCOCO family of datasets.
>     - For SFT data, we conduct the same process as Text4Seg. Specifically, we combine the train split of refCLEF, refCOCO, refCOCO+, and refCOCOg, resulting in a dataset of 800k samples. For RL data, we also use the same four datasets, with ground-truth masks as answers for reward computation.
>
> - The term "web data" was used only in the Pre-training ablation study (Table 3) to show the effect of large-scale pre-training. In fact, the web data is our in-hourse data, we will open-source the data construction pipeline so that developers can annotate their own "web data". Besides, we will release the refcoco data we annotated for reproducibility.
>
> - For model versions, we use GroundingDino-V1.5 for instance detection, SAM2-ViT-H for mask extraction, and Qwen2.5-VL-72B for instance caption. The pipeline will be open sourced.
>
> We hope we have made the data details clear and addressed your confusion. Please raise any unclear points if there still exists.
>
> [1] Lan M, Chen C, Zhou Y, et al. Text4seg: Reimagining image segmentation as text generation[J]. arXiv preprint arXiv:2410.09855, 2024.

---

> ### Author Response · Authors · 2025-11-25
>
> **Q.3 Training Details and Costs**
> - Sequence of Training: 1. Pre-Train (optional), 2. SFT, 3. RL (optional), corresponding to Table 3.
> - Exact meaning of pre-training: Training with large scale web data. The SFT phase is conducted with RefCOCO series datasets.
> - Training Costs:
>     - Pre-training stage was trained for 8,000 iterations, which took approximately 4 days on our setup.
>     - SFT was trained for 1 epoch, which took approximately 2 hours.
>     - RL was trained for 2 epoch, which took approximately 2 days.
>
> We summarize the training details in the table below:
> | Stage              | Pre-train | SFT     | RL      |
> |--------------------|-----------|---------|---------|
> | Data source        | Web data  | RefCOCO | RefCOCO |
> | #samples per epoch | inf       | 800K    | 400K    |
> | Batch size         | 4096      | 256     | 256     |
> | Training iterations    | 8k iter   | 1 epoch      | 2  epoch     |
> | Time consumption     | 4days      | 2h    | 4days   |
>
>
> **Q.4 Clarification of phrases and typos**
> - The term "thin-structure adherence" refers to the model's enhanced capability, primarily gained during the RL phase, to precisely delineate and adhere to the contours of fine and intricate structures (i.e., "thin-structures") within the image.
> - The correct performance metric at 78 tokens, as illustrated in Figure 4, should be 35.6 cIoU, not 53.7 cIoU. We have immediately corrected this mistake in the manuscript and are performing a thorough re-check of the entire paper to ensure no further similar typos remain. We thank the reviewer for pointing out this crucial error.

---

### Official Review · Reviewer_Rubs · 2025-10-29

**Soundness:** 2
**Presentation:** 3
**Contribution:** 3
**Rating:** 4
**Confidence:** 3

**Summary:**

The authors present a two-stage training process that manages to enable MLLMs to achieve significantly improved capabilities on pixel-level perception tasks. Via a combination of SFT and RL, the authors fine-tune models by casting pixel-level problems like segmentation into a sequence-of-point-coordinates format, which effectively allows solving such problems in the language space native to MLLMs, without the need for task-specialised decoder modules.

**Strengths:**

**Originality & Significance:**
- The authors do a great job in outlining why the task matters, as well as how it is approached; all in all a very clear motivation of the presented research
- Simple yet elegant approach that requires no architectural modification of MLLMs to solve (referring expression) segmentation and comprehension
- Good results on the benchmarks, even in the context of architecturally-tailored approaches

 **Quality:**
- The work is placed well within related efforts, and the specific gap the authors tackle is clearly presented and justified
- Appropriate extent of ablation studies to illustrate value/contribution of core components

**Clarity:**
- The work is mostly well written and easy to follow, and a good mix of illustrations and text makes the core concepts and expected results easy to grasp

**Weaknesses:**

- **Justification for use of RL is lacking** and incomplete (imo even misleading), see questions.

- **Several claims are made in a very broad manner** (mainly beyond the focus of this work), and would benefit from being toned-down a bit as they feel overstated (and are not necessarily being substantiated in the paper):
  - E.g. l. 170 *'…can be seamlessly […] integrated as a new, core pre-training task for foundation models'* -> Note that this is quite a big claim, as the authors train/fine-tune a foundation model to solve ONLY this specific task; To substantiate this claim of being a valuable and ‘seamless’ part of pre-training, it would require demonstrating that using this task during pre-training doesn’t result in conflict and/or degradation of other qualities of the model.
  - *'We pioneer the use of RL to optimize the entire generated sequence of points'* (RL is common to refine a sequence of tokens/points); I understand it’s meant in the context of segmentation, but again feels a bit ‘overstated’ (given that there are other works that use RL for segmentation, albeit with different setups & backbones)
  - *'in essence, reinforcement learning is a more reasonable and efficient. Optimization method for perception tasks'* -> Again, this is a very big and broad statement that is not substantiated at this scale; Stating RL as the ‘most efficient’ way of optimising is also a rather surprising statement given the sparse supervisory signal and hence data inefficiency it provides.

  $\Rightarrow$ While I do see what the authors are getting at, such broad statements should only be made if they are substantiated in the paper, and would otherwise benefit from being toned-down a little / from being made more concise and specific


- **Quality/Attention to detail should be improved**, e.g.
  - l. 419/420: ‘[…] moderate density (221 tokens) yields the best score’ -> Does not match Figure 4, where best score is reached at somewhere around 300 tokens
  - incorrect prompts displayed in visualisations: e.g. prompt in Figure 2 lower-right, Fig 9 (‘dragon’ vs. ‘street’),
  - repeated sentences in the text: e.g. l. 157
  - typo in Takeaway2 (‘generalization of on’)
  - Figure 5 should be Table 4

**Questions:**

**TL;DR:** I see the merit of this paper, but have a number of questions/points I’d like the authors to address. I’m happy to reconsider my current rating based on the authors’ response to my questions:

- Justification for RL: “Contours are inherently many-to-one w.r.t. masks; enforcing exact token matching is suboptimal. Reinforcement learning well bridges the gap …”
  $\rightarrow$ I’d like the authors to further elaborate on this aspect, as I think it is currently stated in a very misleading way. To me, the many-to-one nature isn’t actually a problem that’s bridged or solved via RL.
  $\rightarrow$ Rather, the many-to-one mapping is circumvented by using an abstracting metric like IoU, which essentially judges the resulting overlap of the area rather than the point-sequence itself; which is, however, independent from RL.
  $\rightarrow$ The MSE distance between the centroids used as second metric in the RL reward could directly be used as a supervised loss, again circumventing any direct point-sequence judgement;
  $\rightarrow$ In my opinion, the main benefit of RL is that it solves the challenge of backpropagating gradients to the points when using the IoU, which isn’t trivial when trying to use ‘normal’ supervision given that the output is in the language space and therefore discretised/sampled; so RL bridges this disconnect, but not any many-to-one assignment problem; correct?

- L. 255: What exactly is meant by “instantiate queries as Cartesian product of the available elements”?

- Are the examples in Figure 2 results from the training or validation/test set?

- Multiple questions re: Table 3:
  - What do the improvements (in parentheses) each actually refer to, i.e. what’s the baseline for each of these improvements?
  - L406ff: Benefit through pre-training; I can see that SFT benefits through pre-training by the stated 4.6 points (going from 65.5 to 70.1), but SFT+RL goes from 75.2 to 78.5 – which is far from the stated ’13.0’ improvement (rather 3.3); Or am I misinterpreting these?
  - Suggestion (optional): I’d recommend swapping the colours around, so that improvements are highlighted in ‘green’ and decrease is represented in ‘red’ (aligns more with the common association to ‘good’ and ‘bad’)

---

> ### Author Response · Authors · 2025-11-26
>
> We would like to express our heartfelt gratitude for the valuable feedback you've provided on our manuscript. Your in-depth analysis and suggestions are of great significance to us, and we are committed to using them to enhance the quality of our work.
>
> **Q.1 Overstate Claims**
>
> We acknowledge the reviewer's point that some of our claims may have been overstated. We will tone-down and explain some of the claims.
>
> - **Claim on Pre-training**: Our intention was to highlight that the task structure is compatible and modular enough to be incorporated into a large foundation model's pre-training pipeline, should future computational resources allow for such a large-scale integration. We have revised the claim to emphasize its potential for integration rather than asserting its immediate status as a core pre-training task.
>     - We have conducted experiments that integrate the segmentation data into a general visual language data recipe. The results are shown in the table below. Due to time and resource constraints, we have not yet fine-tuned the recipe and training iterations to the optimal, as the SFT phase of VLM requires a significant amount of training cost, but it has achieved baseline performance on general VQA benchmarks and effectively demonstrates the practical compatibility and effectiveness of our SimpleSeg for training a vision-language generalist.
>
> |MMStar|MMBench|AI2D|MMMU|
> |---|---|---|---|
> |48.0|67.7|66.3|35.9|
>
> - **Claim on RL in Segmentation**: We will modify the text to clarify that our novelty lies in applying RL for decoder-free segmentation MLLMs. We provide more discussion in paragraph Q.2 below.
>
> **Q.2 Elaboration on Justification for RL**
>
> The reviewer has correctly identified an ambiguity in our justification for using RL related to the "many-to-one" nature of contours and masks. We deeply apologize for the unclear phrasing. The "many-to-one" nature we intended to convey is:
>   - A single ground truth mask can be accurately represented by multiple valid point sequences (contours), varying across different sampling algorithms and hyperparameters.
>   - In SFT, the model is forced to fit one specific instance of ground truth point sequences sampled during data annotation. This is suboptimal because enforcing an exact match to this single sequence is not the ultimate optimization goal (which is the mask area/overlap).
>
> The reviewer's analysis of RL's core benefit is correct: **The main benefit of RL is that it provides a sequence-level, global supervisory signal (e.g., IoU) which is otherwise challenging to backpropagate directly to the generated points in the discrete language space of the SFT model.**
> This sequence-level optimization allows the model to:
>   - Circumvent the rigid point-sequence matching of SFT.
>   - Adaptively adjust points to compactly and globally optimize the final perceptual metric (like IoU), which directly aligns with the true goal.
>
> Therefore, we have clarified the text to reflect that RL bridges the disconnect between the discrete language space output and the desired continuous, sequence-level optimization metric (like IoU), rather than solving a many-to-one assignment problem.
>
> **Q.3 Typos**
>
> Thank you for pointing out the detailed writing typos and inconsistent presentations. We sincerely apologize for these inadvertent errors. We will meticulously review and revise every detail to enhance the readability of the text.
>
> **Q.4 Clarification on Cartesian Product**
>
> An object has four attributes: `[text, point, bbox, mask]`. A query is a pair of two elements. For example, `(text->mask)` means query the mask given text. The Cartesian Product is the set of a total of 4x4=16 types of queries. Of course, we will omit some trivial queries, such as `(mask->box)`, since they can be implemented with min-max operations.
> We will revise the manuscript to make this definition clearer.
>
> **Q.5 Source of examples in Figure 2**
>
> The examples shown in Figure 2 are results from the test set. We will add a caption to clarify this point.
>
> **Q.6 Clarification for Table 3**
>
> We deeply apologize for any confusion caused. We provide clear clarification and will add an explanation for the settings in the caption.
>
> The first line (only SFT) is the baseline, and all the improvements (in parentheses) actually refer to it, since all experiments contain the SFT phase and can share the same baseline.
> The improvements in parentheses all refer to the first line. That is to say, SFT+RL can bring 9.7 improvements, and pretrain+SFT+RL can bring 13.0 improvements, both compared to pure SFT.
>
> We sincerely appreciate the suggestion regarding the color scheme. We agree that aligning improvements with 'green' and decreases with 'red' will enhance clarity and conform to common visual associations and will update Table 3 accordingly in the revised manuscript.

---

### Official Review · Reviewer_yrN2 · 2025-10-31

**Soundness:** 2
**Presentation:** 2
**Contribution:** 3
**Rating:** 6
**Confidence:** 4

**Summary:**

The goal of the paper is to investigate whether a Multimodal Large Language Model (MLLM) can achieve a high-fidelity image segmentation by predicting mask contours as a sequence of points.

Authors formulate the promptable image (object instance) segmentation task as a sequence generation problem, where the input is a text + image prompt and the output is a sequence of points representing the corresponding (object instance) mask contour and show how to apply the SFT-RL training pipeline to train Kimi-VL to solve this problem.

The contributions of this work are as follows:
1) new MLLM training suitable for the promptable image (instance) segmentation, which "naturally unifies points, boxes, and masks under one textual interface";
2) decoder-free, architecture-agnostic SFT-RL image (object instance) segmentation training pipeline, which, as the authors say, is simple, yet shows strong results on referring expression benchmarks;
3) ablations of the training pipeline-related hyperparameters.

**Strengths:**

1) Strong results on segmentation benchmarks.
2) Significance of task (wide range of practical applications)
3) Simplicity (easy to read and follow paper and implementation details, except points mentioned in Weaknesses (Presentation)).

Although I believe the central claim of the paper is not sufficiently supported by the evidence, given that the results can be applied to various practical applications (such as controllable image editing, vision-based tool use, and GUI-grounded agents), I consider the overall contribution of the paper to be valuable to the community.

**Weaknesses:**

1. The central claim (L92-93) "we demonstrate that standard MLLM architectures possess a strong, inherent capacity for fine-grained perception" is not sufficiently supported by evidence, as the approach is validated only on one VLM architecture -- Kimi-VL. This claim might hold for other architectures, but it was not tested in this work.

2. Some components are not presented clearly or explanation is missing (Presentation):
- L268-266: Not clear what "large scale web data" means in this context, how it was collected, and whether it complies with CoE.
- In L266-267 (Sec. 3.2 Training Pipeline), it is stated that training encompasses SFT and RL phases. However, in L400-401 (Sec. 4.3) and Table 3, pre-training without SFT is also mentioned. Not clear what is done in this context.
- In L319: "Training uses 32 NVIDIA GPUs" neither GPU type nor memory is specified.
- Supplementary materials visualizations lack explanations/interpretation. For example, in Figure 9 it is not clear whether text query "Please recognize and segment the dragon in the image." is duplicated twice for two different images intentionally. If so, what authors aimed to demonstrate by this.

3. Ablation studies are focused on hyperparameters (which is useful indeed) but lack experiments on segmentation results (for example, failure case analysis). See questions for more details.

4. Authors state that "Code, data and model are publicly accessible at https://github.com/simpleseganonymous/SimpleSeg." As of Monday, 27 October, the repository contains just two data preprocessing scripts and the README, which states: "We provide the data processing scripts for our SimpleSeg in data.py, which can generate json files in LLaVA conversation formats."

**Questions:**

1. Was a failure case analysis performed (on what objects approach does not perform well)? For example, how does it work if objects have wholes or when prompted to segment objects which does not appear in the image? It would also be interesting to see the distribution of errors between semantic (i.e., not capturing the correct object from the textual description) and geometric (i.e., inaccurate or invalid mask) errors.

2. How pre-training phase mentioned in L400-401 (Sec. 4.3) and Table 3 is done?

3. Why does text grammar differ for different target types L263-267? Points, rectangles and polygons can be represented using [[x, y], ...] grammar.

4. L156-157: sentence repetition. L276: including -> includes?

**Details Of Ethics Concerns:**

L268-266: Not clear what "large scale web data" means in this context, how it was collected, and whether it complies with CoE.

---

> ### Author Response · Authors · 2025-11-27
>
> We sincerely appreciate your valuable feedback on our manuscript. Your insights are extremely helpful and have provided us with clear directions for improvement.
>
> **Q.1 Results on other architecture**
>
> Thanks for your suggestions. We agree with the reviewer that validating our approach on our architecture can solidify our claim. We are conducting experiments on Qwen3-VL architecture, and will update the results and add them to the manuscript.
>
> **Q.2 Clarification for Web data**
>
> First, for all reported results in Table 1 and Table 2 against state-of-the-art methods, our model is trained without the external "web data". This was chosen to align the training setting and ensure a fair comparison with existing methods.
>
> We clarify the term "large-scale web data" used in this context and confirm its compliance with the Code of Ethics (CoE). The primary data used for training are derived from established, public datasets: LAION and Coyo, and we annotate the labels with our pipeline. The total size of our finalized, annotated dataset is approximately 100M samples.
>
> The discussion regarding pre-training with large-scale web data (and its subsequent lower performance on RefCOCO, as mentioned in the original text) was included primarily as a scaling analysis and ablative study. It was intended to demonstrate the model's fundamental capacity and potential for performance improvement if the training data were to be scaled up with more diverse, web-sourced data. We acknowledge that the connection between the scaling study and the main results could have been clearer, and we will clarify it in the revised version of the paper.
>
> **Q.3 GPU type**
>
> We use NVIDIA H800 80G.
>
> **Q.4 Typos in Figure 9**
>
> We sincerely apologize for the error in Figure 9 of the Supplementary Material. The correct text should be "Please recognize and segment the road in the image" and "Please recognize and segment the snake in the image".
> We will meticulously review and revise every detail to enhance the readability of the text.
>
> **Q.5 Discussion about failure cases**
>
> We strongly agree with the reviewer's insightful suggestion that a failure case analysis can provide critical deeper insights into the model's performance and limitations.
> We have updated the supplementary material PDF to include examples of segmentation results where our model did not perform perfectly.
> - For objects exhibiting non-trivial topology, such as a donut, a single-pass polygon prediction often struggles to achieve perfect delineation. We address this challenge by introducing a two-step prediction mechanism to handle segmentation of discontinuous objects. Specifically, the first step predicts the bounding contour of the outer shape, and the second step predicts the inner hole. This strategy, which can be generalized to objects with multiple discontinuous parts, provides a crucial insight: we can explicitly assign attributes (positive or negative) to individual predicted polygon segments to effectively handle the segmentation of objects with more complex topological structures.
> - We acknowledge that there are still extremely challenging scenarios, such as the example of a zebra shown in the updated supplementary material. The strong texture ambiguity and visual complexity (camouflage) in these cases present a significant difficulty and deserve more efforts.
> - Besides, we show more result examples in the Supplementary Material PDF. Our SimpleSeg can inherently handle **Multiple Disjoint Objects**, **Objects with Disconnected Parts**, and **Background Segmentation**. Cases of **Non-existent Objects** are included in the GRefCOCO dataset, and it can recognize the disappearance when using it for training.
>
> **Q.6 How is pretraining phrase done?**
>
> The only difference between pretraining and the SFT phase is the training data. The pre-training phase is conducted using large-scale web data to provide the fundamental capacity of concept recognition and segmentation. The SFT is conducted using the RefCOCO series dataset.
>
> We summarize the training details in the table below:
> | Stage              | Pre-train | SFT     | RL      |
> |--------------------|-----------|---------|---------|
> | Data source        | Web data  | RefCOCO | RefCOCO |
> | #samples per epoch | inf       | 800K    | 400K    |
> | Batch size         | 4096      | 256     | 256     |
> | Training iterations    | 8k iter   | 1 epoch      | 2  epoch     |
> | Time consumption     | 4days      | 2h    | 4days   |
>
> **Q.7 Text grammar difference for different types**
>
> Using distinct, task-specific grammars (e.g., a simple coordinate pair for a point vs. a sequence of coordinates for a polygon) makes the model's output more explicit and intuitive for a given task. This clearer representation simplifies the learning objective during training and interpretation during inference.

---

### Official Review · Reviewer_bNvn · 2025-11-01

**Soundness:** 3
**Presentation:** 3
**Contribution:** 3
**Rating:** 6
**Confidence:** 4

**Summary:**

This paper introduces SimpleSeg, a minimalist framework that enables MLLMs to perform pixel-level segmentation without any specialized decoders. The key idea is to reframe segmentation as a text generation problem, where the model predicts a sequence of 2D points representing an object’s boundary directly in language space. The training pipeline consists of two stages: SFT for basic grounding and RL with an IoU-based reward to refine contour accuracy. The authors demonstrate that standard MLLMs already possess latent low-level perceptual capacity, which can be unlocked through this pipeline.

**Strengths:**

1. Reframing segmentation as point-sequence generation within language space is a clear, novel, and elegant idea that aligns with the unified philosophy of MLLMs.
2. Despite its simplicity, SimpleSeg performs comparably or better than more complex models, demonstrating that architectural minimalism can achieve strong pixel-level perception.

**Weaknesses:**

1. While the concept is elegant, the technical novelty may be seen as moderate; the method largely combines known ideas (polygon prediction + RL optimization).
2. There are also other decoder-free projects, such as GiT[1] and UFO[2].  It would be better to include some discussion.
3. The evaluation focuses mainly on referring expression segmentation benchmarks; more diverse real-world datasets (e.g., COCO panoptic or open-domain segmentation) would strengthen generalization claims.

[1] "Git: Towards generalist vision transformer through universal language interface." EECV, 2024.

[2] "Ufo: A unified approach to fine-grained visual perception via open-ended language interface." NeurIPS, 2025.

**Questions:**

Can the method in the paper be generalized to the segmentation of multiple objects?

---

> ### Author Response · Authors · 2025-11-25
>
> We would like to express our heartfelt gratitude for the valuable feedback you've provided on our manuscript.
>
> **Q.1 Concerns about a. lack of novelty**
>
> - While prior work has explored polygon prediction, these methods are often task-specific and do not scale. Our key contribution is to formulate a generalist contour representation that fluidly translates pixel-level grounding into the VLM's native sequence-to-sequence format. This "scaling up" of point prediction into a general-purpose VLM is a non-trivial step.
> - The involvement of RL is not an arbitrary combination. As analyzed in `Sec 3.2`, segmenting by sequentially predicting contour points is inherently a long-horizon sequential decision-making process. In this setting, the non-differentiable nature of evaluation metrics and sequence-level credit make the RL objective (specifically, optimizing the final shape quality) a more natural and effective training paradigm than simple SFT. The subsequent performance improvement validates this principled choice, showing that our method is a rational, task-driven architectural choice, not a simple component combination.
>
> **Q.2 Discussion of other methods**
>
> - Thank you for your valuable suggestions on introducing more methods into the discussion, which provided us with clear directions for improvement. We will add these discussions to the revised version of our manuscript.
> - GiT [1] uses a Vision Transformer (ViT) backbone and tokenizes text input via simple word embeddings to achieve open-vocabulary segmentation. Crucially, GiT is built upon ViT. It relies on an architecture customized for visual tasks, limiting its generalization power compared to a true Vision-Language Model (VLM). It still necessitates complex vocabulary and out-of-vocabulary representation designs to handle the task interface, lacking the inherent flexibility and universal prompt-following capabilities of an MLLM.
> - UFO [2] proposes an extra, special mask token and relies on hacking the intermediate feature for retrieval mechanisms. This process is typically performed at a reduced resolution (e.g., $32 \times 32$), which is suboptimal for high-fidelity segmentation. UFO propose to predict multiple mask tokens, which further increases the complexity of the training objective and the problem of low input resolution cannot be solved at its root.
> - PolyFormer [3] is another work, whose idea is also to represent masks with polygons, and we conduct deep comparison with it in response to Reviewer PQGm, referring to https://openreview.net/forum?id=CcQFKeUK5K&noteId=2A2Pi3pF9l.
> - Our SimpleSeg does not need any architecture modification and representation hack. Our work validates the fine-grained visual perception capabilities of the VLM through a more generalized training approach. This success demonstrates that our method can be effectively and scalably integrated into the training pipelines of general-purpose Vision-Language Models.
>
>
> **Q.3 Results on more diverse datasets and multiple objects**
>
> Thanks for your suggestions. We agree with the reviewer that broader evaluation is essential to confirm our generalization claims. We examine our model on more diverse datasets, including COCO Panoptic and RRSIS-D, which is for stuff segmentation and remote sensing images.
>
> Meanwhile, we demonstrate that our model can perform segmentations of multiple objects.
> We provide the results in the Supplementary Material PDF file. It can be seen that our model generalizes well on more extensive data, such as **background stuff segmentation**, **remote sensing segmentation**, **multiple objects segmentation**, and **multi-parts object segmentation**.
>
>
> **Reference**
>
> [1] Wang H, Tang H, Jiang L, et al. Git: Towards generalist vision transformer through universal language interface[C]//European Conference on Computer Vision. Cham: Springer Nature Switzerland, 2024: 55-73.
>
> [2] Tang H, Xie C, Wang H, et al. Ufo: A unified approach to fine-grained visual perception via open-ended language interface[J]. arXiv preprint arXiv:2503.01342, 2025.
>
> [3] Liu J, Ding H, Cai Z, et al. Polyformer: Referring image segmentation as sequential polygon generation[C]//Proceedings of the IEEE/CVF conference on computer vision and pattern recognition. 2023: 18653-18663.
>
> [4] Lai X, Tian Z, Chen Y. Lisa: Reasoning segmentation via large language model. arXiv preprint arXiv: 230800692[J]. 2023.
>
> [5] Yuan Z, Mou L, Hua Y, et al. Rrsis: Referring remote sensing image segmentation[J]. arXiv preprint arXiv:2306.08625, 2023.
>
> [6] Li K, Xin Z, Pang L, et al. Segearth-r1: Geospatial pixel reasoning via large language model[J]. arXiv preprint arXiv:2504.09644, 2025.

---

### Author Response · Authors · 2025-12-02

**Dear Area Chair and Reviewers,**

We sincerely thank the Area Chair and all reviewers for their insightful feedback and stimulating discussions, which have helped us clarify the novelty and core contributions of our work. We summarize the key resolutions below:

**1. Clarification on Novelty and Core Objectives**
Regarding concerns about the novelty of our vertex-based prediction approach:

- **Methodological Advancement:** While polygon-based methods exist, our contribution lies in significantly **scaling up** the number of vertices to represent dense, high-fidelity contours rather than coarse polygons. This transition from simplified shape approximation to fine-grained perception is non-trivial in the context of VLMs. Furthermore, we are the first to successfully integrate **RL** into this paradigm to optimize vertex placement, with solid theoretical consistency.

- **Primary Objective:** We emphasize that segmentation is not the ultimate end, but a means to demonstrate the Pixel-Level Perception Capability of VLMs. As shown in Figure 2 (e.g., line graph parsing, cartoon scenes), our model demonstrates emergent capabilities on out-of-distribution data not present in our training set. This proves that our method successfully leverages and grounds the inherent semantic knowledge of the VLM. **The core target goal of our work is, by achieving strong performance across dense perception task like segmentation, we prove that the VLM inherently possesses/can possess pixel-level perception, rather than merely developing a new segmentation technique or "chasing benchmark scores."**

**2. Robustness and Generalization (Complex Topologies)**
Addressing inquiries regarding generalization (e.g., multi-object and complex topologies like "donuts"):

- We have included extensive qualitative results in the updated supplementary PDF. These demonstrate our method's effectiveness in challenging scenarios, including **Background Stuff Segmentation, Remote Sensing Scenes, Multiple Object Instances, and Multi-Part Objects**.
- We also provided a transparent failure case analysis to outline current boundaries and offer insights into future optimization directions.

**3. Extensive Experimental Validation**
To further validate the universality of our approach, we conducted two major sets of additional experiments:

- **Architectural Generalization:** We verified our method on the Qwen3-VL architecture, demonstrating consistent performance gains. These results will be included in the final manuscript.

- **Generalist Capabilities:** We integrated our task data into a standard SFT pipeline. The resulting model achieves a strong baseline on our segmentation tasks while retaining generic competencies in general QA and disciplinary benchmarks. This paves the way for  and we will continuously make efforts on building stronger vision-language generalists with more resources investment.

**4. Transparency in Training and Data**

- In response to requests for greater reproducibility, we provided comprehensive quantitative specifications and qualitative descriptions regarding our data composition and training protocols in our replies and in the revised manuscipt.

---

### Meta-Review · Area_Chair_PFcK · 2025-12-28

**Summary:**

This submission received diverse scores (66422). The majors concerns (after rebuttal ) from the negative side remains. One primary concern is about the insufficient justification of the proposed approach and implementation details of data construction. There is no evidence to show the reviewers would have changed their score if they had been able to participate fully in the discussion.. As the authors investigated an important and interesting question, I suggest them to carefully consider the concerns raised by reviewers and submit their work to the next conference.

**Reviewer Concerns:**

The authors provided responses to the concerns raised by the reviewers while the reviewers who assigned negative scores did not provide further discussion. T

**Reviewer Scores:**

There is no evidence to show the reviewers would have changed their score if they had been able to participate fully in the discussion.

---

### Decision · Program_Chairs · 2026-01-26

Reject